# Immunogenicity and Reactogenicity of mRNA BNT162b2 COVID-19 Vaccine among Thai Adolescents with Chronic Diseases

**DOI:** 10.3390/vaccines10060871

**Published:** 2022-05-29

**Authors:** Napaporn Chantasrisawad, Thanyawee Puthanakit, Auchara Tangsathapornpong, Chonnamet Techasaensiri, Wanatpreeya Phongsamart, Detchvijitr Suwanpakdee, Peera Jaruampornpan, Jiratchaya Sophonphan, Piyarat Suntarattiwong, Tawee Chotpitayasunondh

**Affiliations:** 1Department of Pediatrics, Faculty of Medicine, Chulalongkorn University, Bangkok 10330, Thailand; napaporn.cha@chula.ac.th; 2Center of Excellence for Pediatric Infectious Diseases and Vaccines, Faculty of Medicine, Chulalongkorn University, Bangkok 10330, Thailand; jiratchaya.w@hivnat.org; 3Thai Red Cross Emerging Infectious Diseases Clinical Center (TRC-EID), King Chulalongkorn Memorial Hospital, Bangkok 10330, Thailand; 4Department of Pediatrics, Division of Infectious Diseases, Faculty of Medicine, Thammasat University, Bangkok 12120, Thailand; auchara@tu.ac.th; 5Department of Pediatrics, Faculty of Medicine Ramathibodi Hospital, Mahidol University, Bangkok 10400, Thailand; chonnamet.tec@mahidol.ac.th; 6Department of Pediatrics, Division of Infectious Diseases, Faculty of Medicine, Siriraj Hospital, Mahidol University, Bangkok 10700, Thailand; wanatpreeya.pho@mahidol.ac.th; 7Pediatric Infectious Disease Unit, Phramongkutklao Hospital, Bangkok 10400, Thailand; detch21@pcm.ac.th; 8Virology and Cell Technology Research Team, National Center for Genetic Engineering and Biotechnology (BIOTEC), National Science and Technology Development Agency (NSTDA), Pathum Thani 12120, Thailand; peera.jar@biotec.or.th; 9Queen Sirikit National Institute of Child Health, Bangkok 10400, Thailand; dr.piyarat.suntarattiwong@gmail.com (P.S.); ctawee@health.moph.go.th (T.C.)

**Keywords:** BNT162b2, immunocompromised, SARS-CoV-2 antibody, adolescent

## Abstract

Adolescents with underlying diseases are at risk of severe COVID-19. The immune response of BNT162b2 may be poor among immunocompromised adolescents. We aim to describe immunogenicity of mRNA BNT162b2 among adolescents who are immunocompromised or have chronic diseases. We recruited adolescents 12–18 years of age; group A impaired-immunity (post-transplantation, cancer, on immunosuppressive drugs) and group B chronic diseases. A two-dose regimen of BNT162b2 was given. Immunogenicity was determined by surrogate virus neutralization test (sVNT) and IgG against receptor-binding domain (RBD). From August to October 2021, 312 adolescents, with a median age (IQR) of 15 years (13.7–16.5), were enrolled (group A 100, group B 212). The geometric means (GMs) of sVNT (% inhibition) against Delta strain and anti-RBD IgG (BAU/mL) after the 2nd dose among group A were: post-transplantation recipients 52.9 (95% CI 37.7–74.2) and 233.6 (95% CI 79–690.6); adolescents with cancer 62.3 (95% CI 29.2–133.1) and 214.9(95% CI 34.2–1348.6); and adolescents with other immunosuppressive conditions 66.7 (95% CI 52.4–84.8) and 849.8 (95% CI 393.4–1835.8). In group B were: adolescents living with HIV 98 (95% CI 97.3–98.8) and 3240.3 (95% CI 2699–3890.2), and adolescents with other chronic disease 98.6 (95% CI 98.3–98.9) and 3818.5 (95% CI 3490.4–4177.4). At day 90, immunity declined; among impaired-immunity participants were 43.9 (95% CI 30.8–62.4) and 178.7 (95% CI 91.2–350.1) and adolescents with chronic diseases were 90.6 (95% CI 88.4–92.8) and 1037.1 (95% CI 933.3–1152.5). In conclusion, adolescents with impaired immunity had a poor response to 2-doses of BNT162b2, additional dose should be considered. Adolescents with chronic diseases had excellent response but immunity waned after 3 m, booster dose may be required.

## 1. Introduction

As of March 2022, more than 430 million people have been infected with severe acute respiratory syndrome coronavirus-2 (SARS-CoV-2) and more than 5.9 million people have died from Coronavirus Disease 2019 (COVID-19), globally [1]. The proportion of children and adolescents, under 20 years of age infected with COVID-19 makes up 18% of these total cases [2]. In the United States (US), COVID-19 cases have been reported in more than 12 million children under 18 years of age representing 19% of total cases [3]. In general, children and adolescents with SARS-CoV-2 infection have milder symptoms compared to adults and those in the elderly population. One study found 20% of children infected with COVID-19 had asymptomatic disease [4,5]. However, some children can go on to develop a life-threatening multisystem inflammatory syndrome in children (MIS-C) which can lead to hospitalization and death [6]. Additionally, the pandemic significantly disrupted learning and caused a deterioration in the mental health of adolescents during school closure and COVID-19 lockdown [7,8].

Vaccination is an important tool along with public health mitigation measures, such as mask wearing, physical distancing, and disease education. The BNT162b2 mRNA vaccine is the first U.S. Food and Drug Administration authorized COVID-19 vaccine for emergency use in adolescents aged 12–18 years old on 10 May 2021 [9], containing nucleoside-modified messenger RNA (mRNA) encoding the SARS-CoV-2 spike glycoprotein [10]. A randomized controlled trial studies in healthy adolescents aged 12–15 years shown high neutralizing titers; 1.76 times higher antibody compared with young adult, with a favorable safety profile and 100% observed vaccine efficacy against COVID-19 from 7 d after vaccination. However, there were limited data among children with chronic diseases especially immunocompromised children [10]. Adults with an immunocompromising disorder or who are receiving treatment with immunosuppressive therapy displayed a reduced immune response following administration of a COVID-19 vaccination [11,12,13,14]. In August 2021, the advisory committee on immunization practices (ACIP) in the US recommended an additional primary dose for moderately or severely immunocompromised people [15]. The Thai Ministry of Public Health devised a policy that prioritizes vaccination for adolescents with high risk of severe COVID-19, including immunosuppressive state or chronic diseases, such as HIV, obesity, chronic lung diseases, and heart disease. These adolescents have a higher risk of severe COVID-19 infection than healthy children [16] and this approach is similar to other vaccination programs, such as in the United Kingdom [17].

There are a variety of surrogate markers for vaccine-induced immune measurement, binding antibody, and neutralizing antibody. Spike receptor binding domain antibody IgG (anti-RBD IgG) is a binding antibody that has a correlation with 80% clinical efficacy—using 506 BAU/mL as a cut-off [18]. The neutralizing antibody represents the protection against infection by blocking the attachment of a virus to the cell. Surrogate virus neutralization test (sVNT) is commonly carried out using an ELISA-based technique, since it is well correlated with both conventional microneutralization assay and pseudovirus-based virus neutralization test [19]. This prospective cohort study is aimed to describe immunogenicity following 2-doses of mRNA BNT162b2 among adolescents aged 12–18 years old in immunocompromised adolescents in comparison with adolescents with other chronic diseases.

## 2. Materials and Methods

### 2.1. Study Design and Participants

This study was a prospective multicenter cohort study conducted at 6 clinical research centers in Thailand: 5 sites in Bangkok (Chulalongkorn University, Queen Sirikit National Institute of Child Health, Phramongkutklao Hospital, Ramathibodi Hospital, and Siriraj Hospital, Mahidol University) and 1 site in Pathum Thani province (Thammasat University). The inclusion criteria were: (1) adolescents, 12–18 years of age; (2) never been diagnosed with COVID-19; and (3) has an underlying medical condition which categorizes into Group A impaired immunity (post-transplantation, cancer, or on immunosuppressive drugs) or Group B chronic diseases (HIV receiving antiviral treatment, heart condition, lung condition, diabetes, or obesity). The exclusion criteria were asymptomatic SARS-CoV-2 defined as a positive result of IgG against receptor-binding domain or antinucleocapsid IgG and receiving lived attenuated vaccines within 4 w.

This study was registered at the Thai Clinical Trials Registry (TCTR20210826001) and approved by the institutional review board from all clinical research sites. At the screening, participants and parents or legal guardians provided written informed assent and consent before enrollment in the study.

### 2.2. Study Procedures

Baseline demographics and clinical data including medical history and current medications were reviewed to determine eligibility against inclusion and exclusion criteria. Participants received two 30 micrograms doses of BNT162b2, (manufactured by Pfizer-BioNTech/Comirnaty^TM^, lot number 30125BA, USA. And FH6387, Belgium) intramuscularly at deltoid muscle 21–28 d apart. After each vaccination, all participants were observed for at least 30 min. The solicited local and systemic reactogenicity (including fever, pain/swelling/erythema at the injection site, headache, malaise, myalgia, arthralgia, vomiting, and diarrhea) was recorded in the diary by participants or their caretakers for 7 d following each vaccination.

Blood samples were obtained from participants at baseline; at 21–28 d following dose 1 of BNT162b2; at 14–28 d following dose 2 of BNT162b2, and at day 90 or 180. Children who had laboratory evidence of previous infection defined as baseline IgG against receptor-binding domain ≥ 50 AU/mL, or positivity for nucleocapsid antibody to SARS-CoV-2 will be excluded from immunogenicity outcome analysis.

### 2.3. Immunogenicity Measurement

Neutralizing antibody against SARS-CoV-2 was measured by surrogate viral neutralizing titer (sVNT) to detect neutralizing antibodies against both the ancestral strain and Delta variant. sVNT is designed to detect neutralizing antibodies targeting the receptor-binding domain of SARS-CoV-2 spike protein by enzyme-linked immunosorbent assay (ELISA) [19], which was performed at the National Center for Genetic Engineering and Biotechnology (BIOTEC), Bangkok, Thailand. Briefly, purified recombinant human ACE2 (hACE2) and the RBD of the spike protein of SARS-CoV-2, including the ancestral strain and Delta strain, were used. Diluted sera were mixed with conjugated horseradish peroxidase (HRP)-RBD. Then, the mixture was incubated and added to 96-well plates coated with 0.1 microgram of recombinant hACE2 ectodomain per well (GenScript^TM^) to perform ELISA. The neutralizing antibody level was detected and reported as per cent signal inhibition (% inhibition) using the following equation:


(1)
$$\%\;inhibition=100\times [1-\frac{sampleOD450}{negativeOD450}]$$


The % inhibition of sVNT, to either the ancestral strain or Delta strain, of ≥ 80% inhibition was considered a threshold level for our study to classify the achievement of 80% protection against infection.

Immune response against SARS-CoV-2 was evaluated by measuring a binding antibody: anti-RBD IgG by Quant assay using Architect system (Abbott diagnostics, Chicago, IL, USA). It is an automated chemiluminescent microparticle immunoassay (CMIA) giving results in AU/mL unit. Samples with a result value above 40,000 AU/mL were automatically diluted and re-evaluated using the dilution factor following the manufacturer’s protocol. The anti-RBD IgG level as AU/mL was converted to binding-antibody units (BAU/mL), WHO International Standard concentration, using the equation BAU/mL = 0.142 × AU/mL [20]. An anti-RBD IgG ≥ 506 BAU/mL was considered as a protective antibody level, which was reported to be associated with 80% vaccine efficacy against primary symptomatic COVID-19 [18].

### 2.4. Reactogenicity

The solicited local and systemic reactogenicity within 7 d following each vaccination was recorded in the diary. The level of reactogenicity was graded into 4 levels: grade 0: no symptoms; grade 1: mild symptoms that did not interference with activities of daily life, or a fever with body temperature (BT) 38.0–38.4 °C; grade 2: moderate symptoms with some interference with activities of daily life, or a fever with BT 38.5–38.9 °C; and grade 3: severe symptoms that significantly limited daily activity, or a fever with BT 39.0–40.0 °C [21]. Moreover, we collected data regarding unsolicited/serious adverse events for 21 d after each vaccination.

### 2.5. Statistical Methods

Baseline demographics and clinical characteristics were analyzed using descriptive statistics and presented as numbers and percentages for categorical variables. Continuous variables were presented as the median and interquartile range (IQR). The Wilcoxon rank-sum tests were applied to compare the continuous variables and the Chi-squared test or Fisher exact test were used to compare categorical variables between two groups. *p*-values < 0.05 were considered statistically significant.

The primary endpoint was to compare the proportion of participants with sVNT ≥ 80% inhibition, against the ancestral strain and the Delta strain, across groups post 14–28 d after receiving 2-dose of BNT162b2. The secondary endpoints were to compare the geometric means (GMs), with a 95% confidence interval (95% CI), of anti-RBD IgG and sVNT at 14–28 d post completion of 2-doses of vaccination, at day 90 or 180. Moreover, we also reported the incidence of reactogenicity that occurred within 7 d following BNT162b2 vaccination. During follow up if a participant was diagnosed with COVID-19, or had received an additional dose of the vaccine, the participant data was censored and not included in the immunogenicity analysis. Data were stored in the Research Electronic Data Capture (REDCap version 6.10.8) system. All analyses were performed using STATA version 15.1 (Stata Corp., College Station, TX, USA).

## 3. Results

### 3.1. Study Populations

During 23 August to 1 October 2021, a total of 312 adolescents aged 12–18 years were recruited at 6 sites. In total, 100 adolescents were immunocompromised (Group A) and 212 were living with chronic diseases (Group B). Group A consisted of: post-transplantation recipients (*n* = 42; 23 kidney and 19 hematopoietic stem cell transplantation, HSCT); adolescents with cancer receiving chemotherapy (*n* = 15); and adolescents receiving other immunosuppressive drugs (*n* = 43). Group B consisted of adolescents with HIV (*n* = 46); obesity (*n* = 58); chronic lung disease (*n* = 33); and others (*n* = 75). The median age among participants was 15 years (IQR 13.7–16.5) and 161 participants (51.6%) were female. The baseline characteristics of the participants at enrollment are shown in Table 1. Adolescents living with HIV receiving antiretroviral therapy (ART) had a median (IQR) CD4 lymphocyte count of 607 cells/mL (504–820), and all of these adolescents had plasma HIV RNA < 50 copies/mL. Of the 42 post-transplantation recipients, including 19 HSCTs (median 37 m after HSCT) and 23 kidney transplantation (median 35 m after transplantation), most recipients received immunosuppressive drugs, such as prednisolone (57%) or mycophenolate (45%). Most of the patients who were receiving immunosuppressive drugs were receiving them for autoimmune inflammatory diseases, particularly systemic lupus erythematosus (SLE); these recipients mainly received prednisolone (73%), methotrexate (28%), or mycophenolate (20%).

In total, 11 participants had laboratory evidence of previous infection prior to vaccination. Therefore, in the immunogenicity cohort we included only 301 participants: 97 in group A and 204 in group B. During follow-up, 6 participants prematurely discontinued the study prior to day 42; 1 participant had COVID-19 infection and 5 participants were lost to follow-up. One participant died due to progression of malignancy at day 61 of the study.

During January–March 2022, Omicron variant became the dominant variant in Thailand. In total, 19 participants in group A (impaired immunity) received an additional dose of the COVID-19 vaccine. There were 23 participants (6/97 (6%) in group A and 17/204 (8%) in group B) who, subsequently, contracted SARS-CoV-2. These included 13 symptomatic participants and 10 asymptomatic participants diagnosed by positive nucleocapsid antibodies to SARS-CoV-2 or an increase in anti-RBD IgG at day 180 compared with day 42 or 90. Therefore, the final number of participants who had data at follow up on day 180 was 172 participants (35 in group A and 137 in group B).

### 3.2. Immunogenicity

#### 3.2.1. SARS-CoV-2 Surrogate Virus Neutralization Test (sVNT)

The sVNT was measured at a median (IQR) of 22 d (21–25) after dose 1 of BNT162b2 prior to getting dose 2 of the vaccine. The GMs of sVNT are shown in Table 2 and Figure 1. The GMs of sVNT against Delta strains were 22.7% inhibition (95% CI 16.9–30.4) and 60.1% inhibition (95% CI 56.9–63.5) among adolescents with impaired immunity (group A) and with chronic diseases (group B), respectively. The geometric mean ratio (GMR) was 0.38 (95% CI 0.31–0.46).

All adolescents with chronic diseases achieved sVNT ≥ 80% inhibition against both ancestral and Delta strains after receiving 2-dose of BNT162b2. However, among adolescents with impaired immunity only 61% (*p*-value < 0.01) and 60% (*p*-value < 0.01) of participants achieved sVNT ≥ 80% inhibition against ancestral and Delta strains, respectively. In this group, 54% of post-transplantation recipients, 53% of cancer patients and 68% of patients with immunosuppressive drugs achieved sVNT ≥ 80% inhibition against the Delta strain which was a predominant strain circulating during August to December 2021 while the study was conducted in Thailand. The GMs of sVNT against the Delta strain after receiving 2-doses of BNT162b2 in adolescents with impaired immunity was 59.7% inhibition (95% CI 49.2–72.6). The GMR was only 0.61 (95% CI 0.54–0.69) compared to adolescents with other chronic diseases. Likewise, the GMs of sVNT against ancestral strains was 62.4% inhibition (95% CI 53.3–73.1), with GMR 0.63 (95% CI 0.57–0.70) compared with adolescents with chronic diseases.

There was wide heterogeneity among adolescents in the immunocompromised group from types of transplantation and types of immunosuppressive agent. Among 23 adolescents with a kidney transplant, all received prednisolone and only 6 (27%) had sVNT against Delta variant ≥ 80% inhibition. In contrast, 16/19 (84%) of HSCT recipients achieved this level where only 7 HSCT recipients were receiving immunosuppressants. Regarding types of immunosuppressive drugs in kidney transplant recipients, 3 of 18 (17%) who were on prednisolone with mycophenolate could achieved this level while 6 of 22 (27%) who were on prednisolone without mycophenolate achieved this level. (*p*-value 0.48)

Among the 38 participants receiving immunosuppressive drugs who attend the post vaccination visit, 27 participants were receiving glucocorticoids and only 59% of them had sVNT against Delta variant ≥ 80% inhibition. Whereas among nonglucocorticoid users in this group of 38 participants, only 1 participant who was receiving mycophenolate, tacrolimus, and hydroxychloroquine for SLE failed this goal.

At visit day 90 after dose 1 of vaccine, which occurred at a median of 103 d (IQR 94–113 d), sVNT had rapidly declined (Table 2). The GMs of sVNT against the ancestral strain among group A (immunosuppression) was 44.7% inhibition (95% CI 33.9–58.9) vs. 94.5% inhibition (95% CI 93.0–96.0) among group B (chronic diseases), respectively (GMR 0.47; 95% CI 0.40–0.57, *p*-value < 0.001). The GMs of sVNT against the Delta strain were 43.9% inhibition (95% CI 30.8–62.4) and 90.6% inhibition (95% CI 88.4–92.8), respectively (GMR 0.48; 95% CI 0.39–0.60, *p*-value < 0.001). Only 49% of participants in group A and 85% of group B maintained sVNT ≥ 80% inhibition against the Delta strain.

At visit day 180 after dose 1 of the vaccine, which occurred at a median of 188 d (IQR 182–194 d), 172 participants visited the clinic and were eligible for analysis (group A 35, group B 137). The GMs of sVNT against ancestral and Delta strains among group B chronic diseases were 67.6% inhibition (95% CI 62.4–73.2) and 64.3% inhibition (95% CI 59.4–69.6), respectively. Among immunocompromised participants, group A, 19 participants got an additional vaccination so were subsequently censored (7 post-transplantation recipients, 2 cancer patients, and 10 patients with immunosuppressive drugs). Among 35 participants who had blood drawn at day 180, the GMs of sVNT against ancestral and Delta strains declined from 92.5% inhibition (95% CI 87.5–97.8) and 90.7% inhibition (95% CI 85.3–96.4) at day 42, to 58.8% inhibition (95% CI 43.6–79.2) and 55.6% inhibition (95% CI 42.4–72.9) at day 180.

#### 3.2.2. Quantitative IgG against Receptor-Binding Domain (Anti-RBD IgG)

Data of anti RBD-IgG are shown in Table 3. The GMs of anti-RBD IgG among adolescents with impaired immunity following receiving 2-dose of BNT162b2 were only, 385.2 BAU/mL (95% CI 205.5–722.2). Whereas the GMs of anti-RBD IgG among adolescents with chronic diseases group following receiving 2-doses of BNT162b2 were 3683.7 BAU/mL (95% CI 3398.5–3992.8). Among the subgroup of immunocompromised adolescents (group A), the lowest GMs were among cancer patients; 214.9 BAU/mL (95% CI 34.2–1348.6), with 7 having ongoing chemotherapy and 2 having the last chemotherapy within 6 m. This was followed by post transplantation recipients; 233.6 BAU/mL (95% CI 79–690.6) where the 23 kidney transplant recipients had GMs of 25.3 BAU/mL (95% CI 5.9–107.3); the 19 HSCT recipients had GMs of 3064.2 BAU/mL (95% CI 1999.6–4695.6); and the patients who received immunosuppressive drugs; 849.8 BAU/mL (95% CI 393.4–1835.8). (As shown in Figure 2)

At visit day 90 after dose 1 of vaccine (median day 103), the anti-RBD IgG level of all participants were declining. The GMs of anti-RBD IgG had declined 2.2-fold to 178.7 BAU/mL (95% CI 91.2–350.1) in adolescents who had impaired immunity and 3.6-fold to 1037.1 BAU/mL (95% CI 933.3–1152.5) in chronic diseases.

At visit day 180 after dose 1 of vaccine (median day 188), the GMs of anti-RBD IgG among adolescents with chronic diseases group declined 11.6-fold to 317.2 BAU/mL (95% CI 283.7–354.6). Among the 35 impaired-immunity participants who had blood drawn at day 180, the GMs of anti-RBD IgG had declined 9.9-fold from 2382.9 BAU/mL (95% CI 1621.9–3500.9) at day 42 to 241 BAU/mL (95% CI 172.7–336.4) at day 180.

### 3.3. Reactogenicity

Overall systemic reactogenicities within 7 d following BNT162b2 vaccination were shown in Figure 3a (after dose 1 of the vaccine) and Figure 3b (after dose 2 of the vaccine). The two most commonly reported systemic reactions were myalgia and fatigue. Overall incidence of reactogenicities was lower among immunocompromised patients compared to those with chronic diseases. After dose 1 myalgia was reported in 14% and 32% of the participants respectively (*p*-value 0.01); fatigue was reported in 14% and 25% of the participants, respectively (*p*-value 0.08). After dose 2, myalgia was reported in 19% of immunocompromised participants and 36% of the chronic diseases group (*p*-value 0.002), and fatigue was reported in 20% and 35% of this group (*p*-value 0.01). One serious adverse event was reported, 1 participant died from progression of high-grade glioma at day 61, not related to the vaccine.

The most frequently reported local reaction was pain at injection site reported in 48% (mild 71%, moderate 29%) and 56% (mild 75%, moderate 22%, and severe 3%) of participants following dose 1 and dose 2, respectively. There are significant differences between the immunocompromised and chronic diseases group: pain was reported in 27% of the immunocompromised group and 62% of the chronic diseases group after dose 1 (*p*-value < 0.001) and 37% vs. 65% after dose 2 (*p*-value < 0.001).

## 4. Discussion

This study showed that following receipt of BNT162b2 mRNA vaccine, only half of immunocompromised adolescents achieved neutralizing titer ≥80% inhibition against SARS- CoV-2 Delta strain up to 3 m post vaccination. In contrast, >90% of the adolescent cohort with other chronic diseases, such as diabetes, chronic heart diseases, chronic lung diseases, and HIV (and receiving ART) achieved high immunogenicity response. Additionally, there is a trend of lower responses among adolescents post solid organ transplantation, and cancer patients receiving chemotherapy. This finding supports the ACIPs recommendation that individuals with impaired immunity should consider receiving an additional dose of BNT162b2 at an interval of 4 w following dose 2.

Immunogenicity and effectiveness of COVID-19 vaccines were impaired in adult patients with immunocompromising conditions [11,12,13,14,22,23,24,25]. During the period of Delta strain predominance, the vaccine effectiveness (VE) of 2-dose of mRNA COVID-19 vaccine against hospitalization was 77% in immunocompromised adults compared to 90% in immunocompetent adults [26]. A study among 20,101 adult patients who were immunocompromised revealed that immunogenicity and VE varied across immunocompromised subgroups. The VE was lowest among organ or stem cell transplant patients (59%) compared to rheumatologic or inflammatory disorders (81%) [26]. Our study findings are also coherent with this study finding.

In total, 54% of post-transplantation recipients, 27% of kidney transplant, and 84% of HSCT recipients, reached sVNT ≥ 80% inhibition against Delta strain while only 3 out of 18 (17%) of those receiving prednisolone and mycophenolate achieved this goal. Studies of adult who received solid organ transplantation [14,23] and HSCT [25] found an association between the use of immunosuppressants (glucocorticoid and mycophenolate) and a poor vaccine humoral response and lower seroconversion rate post vaccination. A study by C. Crane, et al. [27] in adolescent kidney transplant recipients receiving immunosuppressive medications, including calcineurin inhibitor, corticosteroid, mycophenolate, and azathioprine, also found that at a median of 45 d following dose 2 of mRNA SARS-CoV-2 vaccine only 52% of participants had detectable SARS-CoV-2 spike protein antibody.

The proportion of patients with cancer that achieved sVNT ≥ 80% inhibition against Delta strain was only 53%. Despite the small group of cancer patients and the variety of cancer types in this study, including hematologic and solid malignancies, these results were similar trend to a previous study in adults. A. Fendler, et al. [28] reported 68% of patients with hematological and solid malignancies had detectable neutralizing antibody against Delta variant after 2-dose of BNT162b2.

Among patient who receiving immunosuppressive drugs, only two-thirds of them could achieved sVNT ≥ 80% inhibition, 59% of whom were receiving glucocorticoids and had sVNT > 80% inhibition. Studies of antibody responses among patients with chronic inflammatory disease and rheumatic and musculoskeletal diseases found that antibody titers after vaccination were lower among those receiving glucocorticoids and mycophenolate but with no clear difference in those receiving methotrexate, azathioprine, or 6-mercaptopurin [11,12,24]. A study by P. Deepak, et al. [11] found following 2 doses of BNT162b2, prednisolone users compared to nonusers had the GMs (at half-maximal dilution) of anti-SARS-CoV-2 S IgG antibodies of 357 and 2190, respectively; and antibody responses in participants were 65% vs. 92%, respectively. Furer, et al. [12] revealed that the percentage of participants with seropositivity after vaccine dose 2 of BNT162b2 was 66% in glucocorticoid users and 60% in mycophenolate mofetil monotherapy users, while in methotrexate monotherapy users this rate was 92%.

In our study, adolescents with chronic diseases including patients living with HIV receiving potent ART achieved ≥80% inhibition against Delta variant. A study in 143 adults who living with HIV and taking ART with GMs of CD4 lymphocyte count 700 cells/mL found that anti-RBD IgG, and neutralized SARS-CoV-2 pseudo-virus were similar in this group compared to the controls, the immunocompetent health-care workers [29]. This finding is compatible with our study that found adolescents living with HIV receiving ART with a median CD4 lymphocyte count 607 cells/mL produced a strong immune response to COVID-19 vaccination. In terms of obesity, a pivotal phase 3 study of BNT162b2 in adults showed similar vaccine efficacy across baseline BMI subgroups [30].

Previous studies have shown that vaccine-induced immunity decays over time [31]. After 3 m, antibody responses post immunization of immunosuppressed groups waned more rapidly than in the chronic diseases group. The GMs of anti-RBD IgG declined 2.2-fold and 3.6-fold among the immunocompromised group and the chronic diseases group, respectively. The GMs of sVNT against Delta strain declined by 15.8% inhibition among the immunosuppressed group and 7.9% inhibition in chronic diseases group. There is a study that showed that BNT162b1 not only stimulated antibody production but also induces CD4 and CD8 lymphocytes that may provide a role in long-lasting memory against SARS-CoV-2 [32]. A study by E. G. Levin, et al. [31] among health-care workers reported a decay of the neutralizing antibody titers after dose 2 of BNT162b2. In contrast with our study, they found that participants with high initial neutralizing antibody levels tended to decay faster of the antibody up to 70 d after dose 2. There is a study evaluating the durability of antibodies following BNT162b2 vaccination in adolescents. The authors found that at 6 m following dose 2, antibody levels declined to level that comparable to the titer after dose 1 [33]. A similar trend was observed in our study, where the GMs of anti-RBD IgG at the 6 m among chronic diseases group, declined 11.6-fold to 317.2 BAU/mL similar to level at 21–28 d after dose 1. The waning immunity made adolescents potentially vulnerable, especially adolescents with underlying diseases at high risk of severe COVID-19.

Local and systemic reactions often occur after vaccination. Immunocompromised patients reported local and systemic reactions approximately two times less than those with chronic diseases. In our study, we observed that pain at injection site occurred in 62–65% after each dose among immunocompetent participants, which was a lower incidence than previous study in adolescent by R. W. Frenck, et al. [10] that reported 79–86% and study in adult by F. P. Polack, et al. [30] that reported 73–83%. Common systemic reactogenicities were myalgia, fatigue, or headache reported by 36%, 35%, and 30% of participants with chronic diseases after dose 2, respectively. These rates of systemic reactogenicity reporting were only half of what had been reported from the pivotal trial of BNT162b2 among adolescents, aged 12–15 years (65–66%) [10], and adults (52–59%) [30]. However, the variation of reaction rate could be due to different populations studied. Reactogenicities are lower among adolescents with impaired immunity compared with adolescent with chronic diseases, and after dose 2 compared with dose 1 of BNT162b2 mRNA vaccine. There is controversy surrounding a potential association between reactogenicity and humoral immunogenicity with the male sex where some studies demonstrate an association [34]. However, other studies could not find an association [35,36]. In our study, there is a trend of adolescents with impaired immunity having less reactogenicity symptoms after receiving BNT162b2 vaccine which in the same direction of lower immune response post vaccination.

The strengths of this study are that we focused on adolescents with chronic diseases and a wide variety of immunocompromised participants, e.g., kidney transplantation, HSCT, cancer, and autoimmune disorders, who are the population at risk of severe COVID-19. Our data provide evidence to support an additional dose for adolescents with impaired immunity. Secondly, we measured both anti-RBD IgG, binding antibody, and neutralizing antibody against Delta strain which was circulating during the study period.

There were some limitations of this study. Firstly, due to variation of immunosuppressive drugs among study participants, we report overall immune response to the vaccine among these group of adolescents, which may be attributed to concomitant immunosuppressive drugs and underlying diseases per se. Secondly, we used an in-house surrogate viral neutralizing antibody rather than conventional or pseudovirus-based virus neutralization test. We adopted 80% inhibition as a cut-off for high efficacy to protect the adolescents from infection. However, our test was set up, as previously described in C. W. Tan, et al. [19], and has been tested and revealed good correlated with both conventional and pseudo-virus-based methods. Finally, we began conducting our study during the circulation of Delta variant, however at day 90 of collecting data Omicron variant started to circulate and becoming the predominant variant during day 180 of the study. Therefore, the infection that occurred had contribution from both waning levels of antibody and immune escape of the Omicron variant.

## 5. Conclusions

Our findings show that adolescents with impaired immunity had a poor immune response after the 2-dose of BNT162b2 vaccination, and require an additional dose as recommended by the Advisory Committee on Immunization Practices [15]. For adolescents with other chronic diseases, 2-dose of BNT162b provide high neutralizing antibody up to 3–6 m post vaccination. However, with the era of Omicron variants, with immune escape, a booster dose should also be considered.

## Figures and Tables

**Figure 1 vaccines-10-00871-f001:**
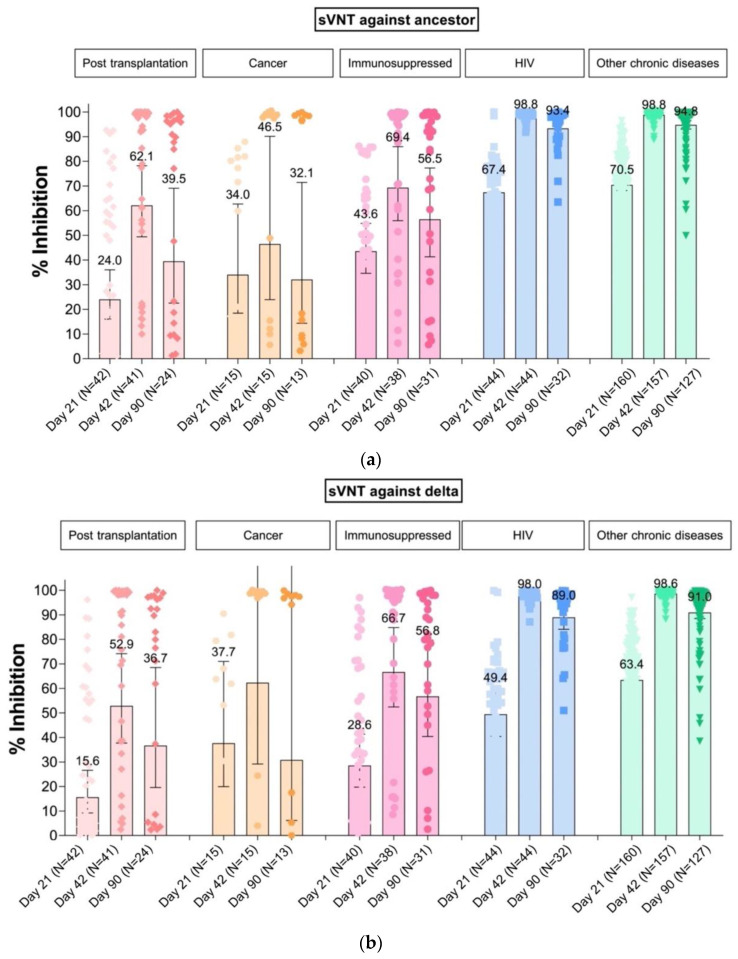
GMs of sVNT among adolescents after BNT162b2 vaccination at day 21, day 42 (14 days after 2-dose of vaccine), and day 90 against the ancestral (**a**) and Delta strain (**b**).

**Figure 2 vaccines-10-00871-f002:**
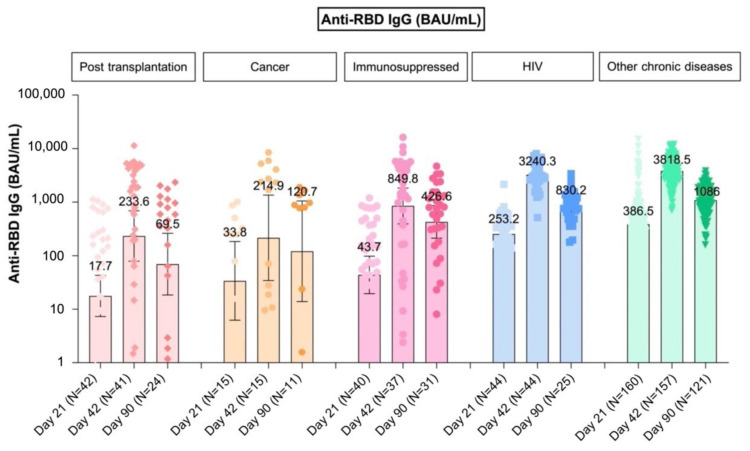
GMs of anti-RBD IgG after BNT162b2 vaccination at day 21, day 42 (14 days after 2-dose of vaccine), and day 90.

**Figure 3 vaccines-10-00871-f003:**
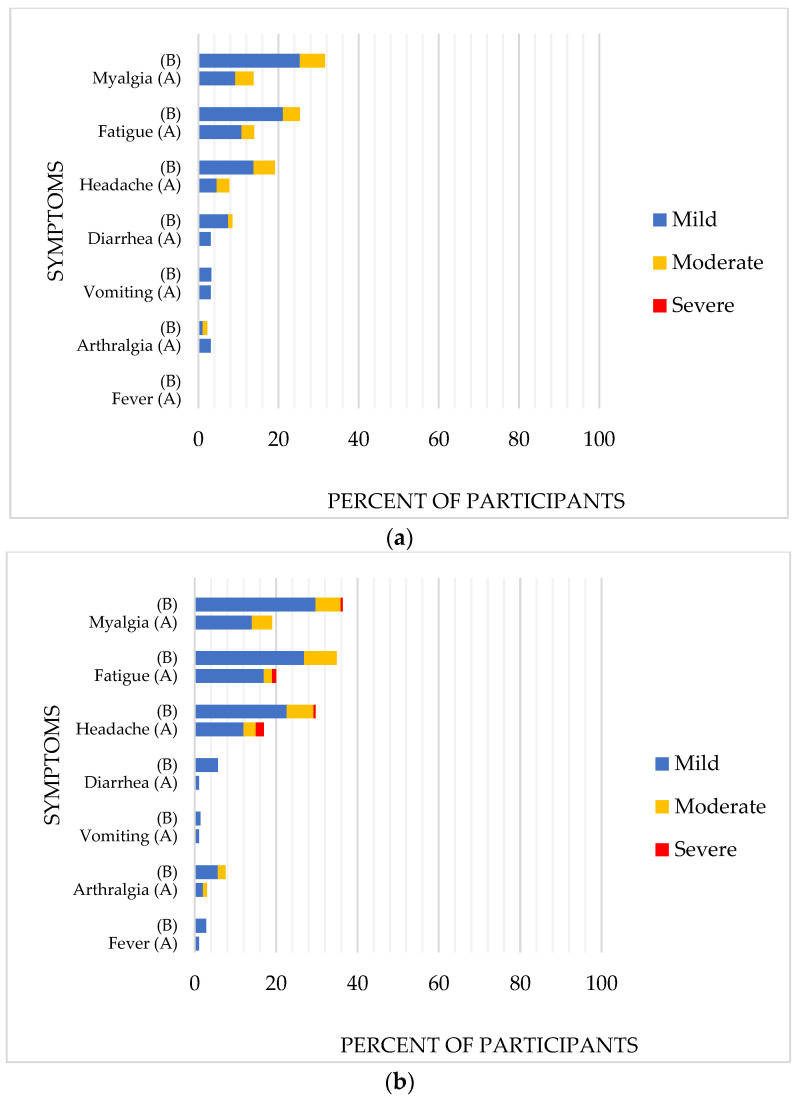
Systemic reactogenicities within 7 d following BNT162b2 among adolescent who are immunocompromised (A), and who have chronic diseases (B); Post dose 1 of BNT162b2 (**a**), Post dose 2 of BNT162b2 (**b**).

**Table 1 vaccines-10-00871-t001:** Baseline characteristics of the study participants.

	Total (*n* = 312)	Immunocompromised (*n* = 100)	Chronic Diseases ^a^ (*n* = 212)	*p*-Value
Age (y),	15	15.5	14.9	0.13
median (IQR)	(13.7–16.5)	(13.9–16.9)	(13.7–16.4)	
Female, N (%)	161 (51.6)	59 (58)	102 (48.6)	0.12
BMI (kg/m^2^), median (IQR)	21.7 (18.1–27.8)	19.7 (17.1–24.2)	22.4 (18.6–30.6)	<0.001
Transplantation, N (%)		42 (42)		
Immunosuppressed, N (%)		43 (43)		
Cancer, N (%)		15 (15)		
HIV, N (%)			46 (21.7)	
Chronic diseases, N (%)			166 (78.3)	

^a^ Chronic diseases included diabetes mellitus, obesity, congenital heart diseases, chronic lung diseases, and chronic kidney diseases.

**Table 2 vaccines-10-00871-t002:** Surrogate viral neutralizing titer (sVNT) against ancestor and Delta strains after BNT162b2 vaccination among adolescents with chronic diseases.

	Post Dose 1		Post Dose 2		Day 90 Post Dose 1	
	Day 21 (*n* = 292) ^a^		Day 42 (*n* = 291) ^b^		(*n* = 227)	
**GMs (95% CI), % inh**	**Ancestor**	**Delta**	**Ancestor**	**Delta**	**Ancestor**	**Delta**
Overall	55.4	45.0	85.7	85.0	75.5	74.1
	(51.1–60.0)	(40.4–50.1)	(81.2–90.5)	(79.8–90.5)	(68.8–82.9)	(66.6–82.4)
Immunocompromised	32.4	22.7	62.4	59.7	44.7	43.9
	(26.0–40.3)	(16.9–30.4)	(53.3–73.1)	(49.2–72.6)	(33.9–58.9)	(30.8–62.4)
- Transplantation (N = 42)	24.0	15.6	62.1	52.9	39.5	36.7
	(16.0–36.0)	(9.2–26.6)	(49.4–78.2)	(37.7–74.2)	(22.5–69.1)	(19.6–68.5)
- Immunosuppressed (N = 40)	43.6	28.6	69.4	66.7	56.5	56.8
	(34.6–54.9)	(19.8–41.3)	(56.0–86.0)	(52.4–84.8)	(41.3–77.3)	(40.4–79.7)
- Cancer (N = 15)	34.0	37.7	46.5	62.3	32.1	30.8
	(18.5–62.7)	(20.0–71.0)	(24.0–90.1)	(29.2–133.1)	(14.4–71.5)	(6.2–154.1)
Chronic diseases	69.8	60.1	98.8	98.5	94.5	90.6
	(67.7–72.0)	(56.9–63.5)	(98.6–99.1)	(98.2–98.8)	(93.0–96.0)	(88.4–92.8)
- HIV (N = 44)	67.4	49.4	98.8	98.0	93.4	89.0
	(62.4–72.8)	(40.4–60.6)	(98.2–99.3)	(97.3–98.8)	(90.0–96.8)	(84.1–94.1)
- Others chronic diseases (N = 160)	70.5	63.4	98.8	98.6	94.8	91.0
	(68.2–72.8)	(60.9–66.0)	(98.6–99.1)	(98.3–98.9)	(93.1–96.5)	(88.5–93.5)
GMR (95% CI) immocompromised vs. chronic diseases	0.46	0.38	0.63	0.61	0.47	0.48
	(0.40–0.54)	(0.31–0.46)	(0.57–0.70)	(0.54–0.69)	(0.40–0.57)	(0.39–0.60)
*p*-value	<0.001	<0.001	<0.001	<0.001	<0.001	<0.001

*p*-value from *t*-test; GMs = Geometric means, GMR = Geometric means ratio, sVNT = surrogate virus neutralization test. Seronegative correspond to sVNT 0% inhibition that were not included. ^a^ Seronegative 9 participants; ^b^ Seronegative 4 participants.

**Table 3 vaccines-10-00871-t003:** Anti-receptor binding domain IgG (Anti-RBD IgG) after BNT162b2 vaccination among adolescents with chronic diseases.

	Post Dose 1	Post Dose 2	Day 90 after Dose 1
	Day 21 (*n* = 299) ^a^	Day 42 (*n* = 294) ^b^	(*n* = 212)
GMs (95% CI), BAU/mL			
Overall	158.6	1803.5	599.9
	(125.0–201.3)	(1423.1–2285.5)	(469.8–765.9)
Immunocompromised	28.5	385.2	178.7
	(16.4–49.5)	(205.5–722.2)	(91.2–350.1)
- Transplantation (N = 42)	17.7	233.6	69.5
	(7.3–42.9)	(79.0–690.6)	(18.4–262.1)
- Immunosuppressed (N = 40)	43.7	849.8	426.6
	(19.6–97.8)	(393.4–1835.8)	(211.9–858.7)
- Cancer (N = 15)	33.8	214.9	120.7
	(6.2–183.6)	(34.2–1348.6)	(13.9–1050.1)
Chronic diseases	352.8	3683.7	1037.1
	(307.0–405.6)	(3398.5–3992.8)	(933.3–1152.5)
- HIV (N = 44)	253.2	3240.3	830.2
	(197.1–325.3)	(2699.0–3890.2)	(617.5–1116.1)
- Others chronic diseases (N = 160)	386.5	3818.5	1086.0
	(328.7–454.5)	(3490.4–4177.4)	(970.9–1214.6)
GMR (95% CI) immocompromised vs. chronic diseases	0.08 (0.05–0.12)	0.10 (0.07–0.16)	0.17 (0.11–0.28)
*p*-value	<0.001	<0.001	<0.001

*p*-value from *t*-test. GMs = Geometric means, GMR = Geometric means ratio, anti-RBD IgG = IgG against receptor-binding domain. ^a^ 2 participants had 0 BAU/mL of anti-RBD IgG; ^b^ 1 participant did not draw enough blood to perform the test.

## Data Availability

The data supporting this study’s findings are available from the corresponding author upon reasonable request.

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
