# Peer review of "Immunogenicity and Reactogenicity of mRNA BNT162b2 COVID-19 Vaccine among Thai Adolescents with Chronic Diseases"

_vaccines, 2022, doi:10.3390/vaccines10060871_

Round 1

Reviewer 1 Report

The paper of Chantasrisawad et al. is well written and gives an important contribution to the field. I have some minor changes to suggest:

- calculate and if any indicate any statistical significance in the figures;
-Do any particular therapy affects antibody levels?
For example, the antiretrovirals used in HIV patients certainly influence the excellent results of this group of samples. Discuss them.
- the authors have to explain why they consider 80% as a percentage of inhibition;
-stratify the drugs used in the immunosuppressive drug group and check for any differences;
- the abstract must be reviewed in the conclusions. Why do the authors write about "fair response"?
- in the limitations should be included authors did not used for neutralizing antibodies the gold standard method ( PRNT) or at least the only FDA-approved sVNT (Genscript -cPass)
- It would be interesting to discuss the symptom data of reactogenicity in comparison with those of adults.

Reviewer 2 Report

Napaporn Chantasrisawad and colleagues present a high quality and well-written experimental manuscript that reports immunogenicity and reactogenicity of mRNA BNT162b2 COVID-19 vaccine among Thai adolescent with chronic diseases.

Authors aimed to describe immunogenicity of mRNA BNT162b2 among adolescents with immunocompromised or with chronic diseases.

Authors recruited adolescents 12-18 years of age; group A impaired-immunity (posttransplantation, cancer, on immunosuppressive drugs) and group B chronic diseases. Two-dose regimen of BNT162b2 were given. Immunogenicity was determined by surrogate virus neutralization test and IgG against receptor-binding domain.

Authors suggest that the strength of their study is that they focused on adolescents with chronic diseases and a wide variety of immunocompromised participants e.g., kidney transplantation, HSCT, cancer, and an autoimmune disorder, which are the population at risk of severe COVID-19. Their data provides evidence to support additional dose for adolescent with impaired immunity. They also measured both anti-RBD IgG, binding antibody, and neutralizing antibody against Delta strain which was circulating during study period.

Finally, authors conclude that adolescent with impaired immunity had fair response to 2-dose BNT162b2, additional dose should be considered. Also, adolescent with chronic diseases had excellent response but waned after 3 months, therefore booster dose may be needed.

Overall, the manuscript is highly valuable for the scientific community and should be accepted for publication.

======================

Other comments to authors:

1) Please check for typos throughout the manuscript.

Author Response

We already have checked and revised typos throughout the manuscript.